# Vaccine Hesitancy and the Green Digital Pass: A Study on Adherence to the Italian COVID-19 Vaccination Campaign

**DOI:** 10.3390/ijerph19052970

**Published:** 2022-03-03

**Authors:** Giuseppina Moccia, Luna Carpinelli, Giulia Savarese, Francesco De Caro

**Affiliations:** Department of Medicine and Surgery, University of Salerno, 84084 Baronissi, Italy; gmoccia@unisa.it (G.M.); lcarpinelli@unisa.it (L.C.); fdecaro@unisa.it (F.D.C.)

**Keywords:** COVID-19, vaccine hesitancy, Green Digital Pass, adherence to the vaccination campaign

## Abstract

Background: In July 2021, the vaccination campaign in Italy suffered a sudden setback, and the number of vaccine administrations decreased dramatically. On 20 July 2021, the obligation of the Green Digital Pass came into force in order to access work and leisure places, penalizing those who had not been vaccinated. The purpose of this work was to investigate the phenomenon of vaccination hesitancy and the underlying reasons, as well as any changes to the membership following the obligation of the Green Pass. Methods: A total of 83 subjects (45.8% F; mean age 22.24 ± 4.308) participated in the survey during the post-vaccine observation phase at the Vaccinal Center of the University Hospital “San Giovanni di Dio e Ruggi d’Aragona” (Salerno, Italy). The questionnaire collected anamnestic information, as well as data on state anxiety (STAI-Y), perception of quality of life (SF-12), perception of COVID-19 risks, and vaccine hesitancy. Results: Among participants, 19.3% reported hesitation. The most common concerns about the COVID-19 vaccine concerned safety and efficacy (4.9%) and the obligation of the Green Pass (4.9%). Conclusions: Findings suggest that delving into the phenomenon of vaccine hesitancy can help to enhance vaccination strategies in order to gain widespread acceptance, a key path to ensuring a quick way out of the current pandemic emergency.

## 1. Introduction

Skepticism in vaccinations is a phenomenon that has existed since the availability of the first vaccine; however, it is currently supported and amplified by the ease with which anyone can find conflicting information on the internet, as well as many other reasons that often have nothing to do with getting vaccines.

The phenomenon defined as *vaccine hesitancy* (a term that includes the concepts of indecision, uncertainty, delay, and retention) is complex and closely linked to various factors with different determinants: historical period, geographical area, and situation [1].

Recognizing the importance that this phenomenon has in the achievement of pre-established health goals, the Strategic Advisory Group of Experts (SAGE) on Immunization of the World Health Organization (WHO), in 2012, created a specific working group on the subject. The material produced was collected and published in August 2015, in a monographic issue of the journal Vaccine, entirely dedicated to vaccine hesitancy, and entitled “WHO Recommendations about Vaccine Hesitancy” [2].

The SAGE stresses that it is urgent and necessary to develop institutional systems and organizational skills at the local and global levels in order to proactively define, monitor, and address vaccination hesitancy, as well as respond promptly to antivaccination movements in the event of misinformation or potential adverse events (adverse events following immunization, AEFI) [3].

Another aspect underlined by the monograph is the urgency to share as much as possible, involving the largest number of stakeholders in the decision-making process on vaccination programs and in the communication process relating to the organization and provision of vaccination services.

With regard to the COVID-19 vaccination campaign, the consequent achievement of adequate vaccination coverage represents, to date, the only winning formula for overcoming the health, economic, and social difficulties brought about by the pandemic, which, since 2019, has gripped countries all over the world. However, there is still resistance to the administration of currently available vaccines for the prevention of SARS-CoV-2 infections, and it is evident that, even in a small group of the population, such resistance could undermine the return to normal [4]. Not surprisingly, the World Health Organization has identified vaccine hesitancy, i.e., the tendency to delay or refuse a vaccine, as the third greatest threat to global health [5]. These fears are also present in healthcare workers [6,7] and school workers [8,9]. The most pronounced concern is the fear of the side effects of the vaccine itself, as well as a strong distrust in pharmaceutical companies due to alleged perceived financial interests and a lack of clear communication on side effects [10]. Numerous variables have been related to this hesitation toward vaccines, including sociodemographic and psychological variables [11]. Reiter et al. [12] surveyed U.S. adults, showing that 69% were willing to receive a COVID-19 vaccine and that this availability was greater in those who reported higher levels of perception of contracting a COVID-19 infection, or who reported being aware of the perceived severity of the infection and the effectiveness of a vaccine.

The updated data in Italy regarding the administration of the anti-SARS-CoV-2 vaccine indicate that 91.12% of the population underwent the first dose of the vaccination cycle and 88.72% underwent the vaccination cycle [13]. In the European Union, 73.7% of the population has had at least one dose of the vaccine, while in the rest of the world the percentage is 59.7% [14].

## 2. The Italian Background

In Italy, a country with a population of about 60 million inhabitants, the vaccination campaign began on 21 December 2020 and, to date, 31,390,566 have completed the vaccination cycle, i.e., 58.12% of the population over 12 [15]. On 30 July, 517,837 doses were administered for the day compared to an average of 550,992 in the previous week, suggesting a decrease by 39,765 daily doses [16].

The Italian vaccination program began with healthcare personnel, as well as people aged 80 and over. Vaccination of younger groups followed progressively but was subsequent to those with frailty (for example, with disabilities). However, some people and antivaccination social groups have contributed to growing anxieties about the vaccination process.

In mid-July 2021, the vaccination campaign in Italy suffered an abrupt halt because the number of people ready to be vaccinated drastically decreased. The Rt contagion index, which was 0.5, in the face of greater freedom from the containment measures given by the government, began to rise, affecting the number of ordinary hospitalizations and those in COVID wards and intensive care (where the number of places available is still limited), as well as that of deaths.

Currently (21 December 2021; Rt contagion index = 4.7), the government has made the Green Pass obligatory for teachers and school staff, health professionals, and for law enforcement. On 1 September, the pass became mandatory for trains and buses, domestic flights, and ferries. The Green Pass obligation in the workplace is backed by the country’s trade unions and the majority of Italian workers; however, it resulted in sporadic protests around the country. As of 20 December, just over 88.62% of Italy’s vaccinable population (over the age of 12) was fully vaccinated against COVID-19 [13,17].

In the European legal context, the reference point (also for Italy) is Regulation (EU) 2021/953 of the European Parliament and of the Council of 14 June 2021 on a framework for the issuance, verification, and acceptance of interoperable COVID-19 vaccination, test, and recovery certificates (EU Digital COVID Certificate) to facilitate free movement during the COVID-19 pandemic.

On 20 July 2021, the Italian Parliament issued legislation on the mandatory nature of the Green Digital Pass (hereinafter referred to as the “Green Pass”), i.e., a document that certifies that one is vaccinated or cured of COVID-19. The Green Pass allows entry to certain places only for those who have been vaccinated, penalizing those who have not. People who have been vaccinated can download the pass from the “IO” app or the website of the Italian Ministry of Health or use a printed document with a QR code. They must show this permit to travel and access public places, such as restaurants and gyms.

COVID-19 certificates should be examined within the overall broader regulatory response to the COVID-19 pandemic, which has been characterized by widespread limitations on different human rights: mobility, curfews, closure of educational institutions, and restrictions of commercial activities. The necessity for the creation of COVID-19 certificates must, therefore, be found in the need to alleviate some of the limitations placed on the general population.

The COVID-19 certificate is to “facilitate safe free movement” in Europe, and it represents a tool for the regulation and governance of the pandemic, as well as for the wider governance and regulation of populations and territories, including the regulation of access to fundamental human rights [18].

Between 21 and 28 July 2021, the press, with an immediate rebound on social media, placed a great emphasis via the news on the Green Pass, and there was a boom in booking for vaccination. In the first week of August 2021, according to the weekly government report published by the extraordinary commissioner for the COVID-19 emergency, the total number of administered doses was 71,071,465, with an increase in the prior week of 3,316,075. The government regulation of 1 August regarding suspension from work without any remuneration or incentive for medical, health, and school/university personnel without a Green Pass also contributed to this figure. Furthermore, the Green Pass became mandatory for students over 12 years of age for school and university attendance. As of 7 August 2021, the measures became operational in various health centers with enormous mass-media prominence. On 9 August 2021, the Italian Health Minister proudly declared that 2 million Green Passes had been downloaded from the ministerial platform in 2 days; and following the new restrictions on 10 December 2021, 1.4 million had been downloaded [13].

In a bibliographic review [19], some of the factors that can influence the acceptance or refusal of vaccination were highlighted, delineating them according to three apparently independent phenomena: (1) low age was associated with a lower willingness to receive vaccination; (2) high concern about being infected increased the likelihood of joining the vaccination campaign; (3) no difference was observed between those who were infected and those who did not get sick with COVID-19. It is important to remember that the perception of risk is an important factor influencing risk behaviors, and thus, people with a low perception of risk tend to reduce prevention.

Furthermore, the most common reasons for vaccination hesitancy in the population were the following: being against vaccines in general; concerns about the effectiveness of the vaccine having been produced in a short time; considering the harmless nature of COVID-19; lack of confidence in the politics of one’s country; and general doubts about the probable existence of the SARS-CoV-2 virus.

It is imperative to have greater analytical skills to identify areas where hesitation is created. For this, the final recommendations of SAGE focus on three main categories: understanding the determinants of vaccine hesitancy; highlighting the organizational aspects that facilitate membership; understanding the tools needed to solve this phenomenon.

On these premises, the purpose of this investigation was to verify the phenomenon of vaccination hesitancy and the reasons behind the refusal of or procrastination with respect to the COVID-19 vaccine and subsequent acceptance, following the mandatory nature of the Digital Green Pass. To this end, the patients of the Vaccinal Center of the University Hospital “San Giovanni di Dio e Ruggi d’Aragona” (Salerno, Campania, Italy) were monitored and interviewed.

## 3. Materials and Methods

### 3.1. Territorial Background and Participants

The sample included 83 people (45.8% women). The mean age was 22.24 ± 4.308 and ranged from 13 to 31 years. The Vaccinal Center covers, for health reasons, the city of Salerno (Campania, Italy) with 128,302 inhabitants (data updated to 2021), and 82.47% took the first dose of the COVID-19 vaccine. However, in order to facilitate the safe continuation of the vaccination campaign and to avoid inconvenience, the “ASL Salerno” (local health institution) has strengthened offers of COVID-19 vaccine administration centers to set up a “HUB”, i.e., temporary health centers used at schools, theaters, churches and municipal buildings, that offer citizens a choice to book the day and location according to their preferences.

### 3.2. Procedure and Data Collection

Our study is characterized by the analysis of data obtained from an online interview carried out at the Vaccinal Center at the University Hospital “San Giovanni di Dio e Ruggi d’Aragona”. Participants represent a “convenience” sample, as they were recruited during the post-vaccine “observation” phase, as required by the health surveillance protocol. Participants were aged from 13 to 31 years and resided in Salerno (Campania, Italy). The survey was performed from 19 to 31 July 2021, following the decision of the Italian Ministry of Health to give subjects of any age group the opportunity to join the COVID-19 vaccination campaign and to make the Green Pass more effective. The data management was performed in accordance with the General Data Protection Regulation of the European Union.

### 3.3. Instruments

The questionnaire was adapted using a survey tool previously applied in our studies [10,20]; the questions focused on anamnestic characteristics and the possible presence of pathologies, as well as on the evaluation of the quality of life and anxiety states, COVID-19-related experiences, perceived risk of infection, and the likelihood of accepting the COVID-19 vaccination. This survey was designed to be completed in approximately 15 min. Specifically, the following standardized scales were used to assess the quality of life and anxiety states:-Short Form-12 [21]: The SF-12 is composed of 12 items (derived from the 36 in the original SF-36 questionnaire) which produce 2 measures relating to 2 different aspects of health: physical health (physical component summary—PCS) and mental health (mental component summary—MCS). The subject is asked to answer on how they feel and on how they manage to carry out typical activities, evaluating the day on which the questionnaire is completed and the previous 4 weeks. This questionnaire has been translated and culturally adapted into various European languages and countries, including Italy, within the IQOLA project [22]. The test–retest correlation after 2 weeks is 0.89 for the index of physical health and 0.76 for the index of mental health.-State-Trait Anxiety Inventory (STAI-Y) [23]: The STAI-Y is divided into 2 scales (Y1 and Y2), each consisting of 20 items, which respectively evaluate state anxiety, through questions relating to how the subject feels at the time of administering the questionnaire, and trait anxiety, with questions that investigate how the subject usually feels. The subject evaluates, on a scale from 1 to 4 (1 = not at all; 4 = very much), how well different statements fit their behavior. For our survey study, only the Y1 scale relating to state anxiety was administered.

### 3.4. Statistical Analysis

Statistical analysis was carried out using IBM software SPSS v. 23.0 (Armonk, NY, USA: IBM Corp.). Both quantitative and qualitative analyses of the scores and items of the SF-12 and STAI-Y tests were conducted on the basis of the frequency of the subjects’ responses and of which we reported the mean, standard deviation (SD), minimum-maximum (Min-Max) and *p* values.

The qualitative interpretation of the means of the scores obtained in the tests was considered with the reference values of the cut-offs for both SF-12 [21] and STAI Y [23].

For items 21–28, a descriptive analysis was carried out through the response frequencies.

## 4. Results

### 4.1. Sample Health Characteristics

The sample reported the presence of chronic diseases (4.8%) and joint pains (4.8%), whereas there were no cases of walking disturbance, hypertension, diabetes, or heart disease. By analyzing the percentage of response frequencies (*p* value < 0.001) to item #1 of the SF-12 “*In general, would you say your health is*” (see Table 1), participants responded “excellent” (21.7%), “very good” (43.4%), “good” (28.9%), and “fair” (6%). All subjects underwent the administration of the first dose of the Pfizer–BioNTech vaccine (at the time of the interview, the only vaccine available at the vaccine center).

### 4.2. Psychological States

The mean scores obtained for the PCS (15.40; SD = 2.012) and MCS (14.19; SD = 1.163) indices of the SF-12 were within the normative range of 0–39, indicating a functional level of perception of one’s health, both physically and mentally. The total mean score obtained for STAI-Y was equal to 39.35 (SD = 9.394; Min = 20, Max = 65) with 30.1% of the sample falling within the category of “mild” with respect to state anxiety (*p* value = 0.041).

Furthermore, to verify the mood of the respondents, we analyzed the response frequencies (*p* value < 0.05) for specific items of both the SF-12 and the STAI-Y (see Table 1 and Table 2). Specifically, for item #9 of the SF-12 “*Have you felt calm and peaceful?*”, 31.3% of the sample answered “most of the time”, whereas for item #11 “*Have you felt downhearted and blue?*”, 42.7% of the sample answered “a little of the time”. Regarding the specific items examined in the STAI-Y, the participants responded for the most part “somewhat” to “*I feel tense*” (43.4%, item #3), “*I feel frightened*” (26.5%, item #9), “*I feel nervous*” (47%, item #12), and “*I feel indecisive*” (22.9%, item #14).

### 4.3. COVID-19 Experience: Risk, Perception, and Prevention

To verify the perception of the situation generated by the pandemic, the risk of contagion from COVID-19, and the implementation of useful behaviors in order to prevent infection, ad hoc questions were constructed according to the CHERRIES criteria [24]. From the analysis of the response frequencies of the participants (*p* value < 0.05), the following emerged: 48.2% reported being personally concerned about the problems related to the pandemic (item #21); 37.3% appeared to be quite worried about being personally and directly affected by the pandemic (item #22); 43.4% were quite worried, considering it likely that their family and friends could be directly affected by the pandemic (item #23); 9.6% were very much in agreement in thinking that they would most likely fall ill with COVID-19 (item #24a); 39.8% were in extreme agreement in stating that getting sick with COVID-19 can be serious (item #24b); 38.6% believed that COVID-19 would affect many people in Italy (item #24c); 3.6% believed COVID-19 to be an invention (item #24d).

Among the behaviors implemented in order to prevent the spread of the infection, the following was revealed: 98.8% of respondents often washed and sanitized their hands (item #25a); 86.7% of respondents maintained their distance from other people (item #25b); 86.7% of respondents always used a mask (item #25c); 97.6% of respondents were vaccinated against COVID-19 (item #25d). In addition, 80.7% stated that they were vaccinated upon receiving the first call from the ministerial platform, while the remaining 19.3% declared the following reasons for not having been initially vaccinated: “*waiting for greater confidence in the efficacy of the vaccine*”, “*having had too little notice*”, “*doing it for the Green Pass*”, “*doing it to be able to travel*” (items #26 and #27).

Furthermore, the reasons that led to the decision to get vaccinated were evaluated (item #28) (*p* value < 0.01), with 85.4% stating that “*Getting vaccinated is the only way to return to our life, to normal*”, 4.9% stating that “*Vaccinating is the only way to get the Green Pass*”, 4.9% stating “*I preferred to wait to get more information on the risks of the vaccine*”, and 1.2% stating “*The information given by the mass media is conflicting*”.

## 5. Discussion

Tools, such as the Digital Green Pass, are useful for achieving overall behavioral change and effects on wellbeing, but they could harm identifiable social groups [25].

Behavioral research has shown that individuals deviate in predictable ways from regulatory policy. For example, default rules often have a major effect on social outcomes as, due to inertia and procrastination, people tend not to make affirmative choices; “framing” and presentation of information are also strategic interventions to influence choices; behavior patterns are also heavily influenced by the emergence of social norms, as people are constrained by reputational forces and concern about the perceptions of others. Moreover, evidence suggests that salient and vivid warnings are more effective than statistical and abstract information [26].

In our case study, the percentage absence in July 2021 was equal to 14% of those called for the first dose of vaccine, compared to values of 3.9% in March, 4.24% in April, 4.34% in May, and 3.57% in June of the same year. According to the data in the possession of the University Hospital, San Giovanni di Dio e Ruggi d’Aragona, there was an evident resumption in reservations on the ministerial platform starting on 19 July, with about 1000 more people than the previous week.

Most scholars agree that confidence and trust play a critical role in reducing vaccine hesitancy across contexts and that the current SARS-CoV-2 pandemic is partially attributed to the decline in trust in science and medicine across the globe.

The results of a study conducted to evaluate the acceptance of COVID-19 vaccination in a sample of elderly people in southern Italy, during the advanced vaccination campaign performed in Italy, show a high percentage of individuals vaccinated or willing to be vaccinated against SARS-CoV-2 (92.7%). However, there was a percentage of individuals who stated general vaccine acceptance was low (45.1%) which was significantly lower than that of people immunized against COVID-19 (86.6%) [27].

Many studies strongly suggest that trust in scientists and domestic healthcare professionals, combined with confidence in the WHO, represents an important driver of vaccine acceptance across the globe. Therefore, for trust and confidence, political leaders should assign resources to the management and communication of vaccine safety, its effectiveness, and distribution protocols to scientists and health professionals. Health professionals should, in turn, participate in developing and deploying communication strategies [28].

Understanding the factors that influence the choice to consent or not to vaccination, allows for targeting of barriers that hinder vaccine uptake [1]. The World Health Organization defines vaccine hesitancy as a “delay in acceptance or refusal of safe vaccines despite the availability of vaccine services” [5,29]. Confidence in the importance (necessity and value) of vaccines has the strongest association with vaccine uptake [29].

Furthermore, it appears from our study that, as opposed to health anxiety or altruism, it was the fear of further social limitation that led to a reduction in vaccine hesitancy.

We believe that these findings will be useful to prevent false beliefs and representations among the population through coherent and simple political and mass-media information. As reported in the literature, the identification of those most likely to exhibit vaccine hesitancy allows for the optimization of available resources and focusing on the communicative effort, thereby increasing the effectiveness of awareness campaigns and attaining a higher immunization rate in less time [10].

## 6. Conclusions

Public health spending is an investment that creates long-lasting health and social impacts that outweigh any initial cost savings. Indeed, we have learned this lesson during the COVID-19 pandemic [26].

Razai et al. [29] summarized the following factors related to interventions increasing vaccination uptake: communication from trusted sources, such as community representatives, healthcare providers, and local authorities that is culturally relevant and accessible in multiple languages; access to vaccines with flexible Green Pass practices and outreach programs via delivery models in the community; community engagement to raise knowledge and awareness of vaccinations; training and education of those involved with engagement activities at a local level.

Despite being a local study with a limited sample size, the data collected enabled us to identify false beliefs and representations conveyed in the messages gathered from media and social networks.

## Figures and Tables

**Table 1 ijerph-19-02970-t001:** Percentage of response frequencies (%) relating to items #1, #9 and #11 of the SF-12.

SF-12					
Item #1 “In general, would you say your health is”					
Excellent	Very good	Good	Fair	Poor	
21.7	43.4	28.9	6.0	-	
Item #9 “Have you felt calm and peaceful?”					
All of the time	Most of the time	A good bit of the time	Some of the time	A little of the time	None of the time
14.5	31.3	19.3	22.9	10.8	1.2
Item #11 “Have you felt downhearted and blue?”					
-	7.3	6.1	29.3	42.7	14.6

**Table 2 ijerph-19-02970-t002:** Percentage of response frequencies (%) relating to items #3, #9, #12 and #14 of STAI-Y.

STAI-Y				
	Not at All	Somewhat	Moderately So	Very Much So
Item #3 “I feel tense”	30.1	43.4	19.3	7.2
Item #9 “I feel frightened”	66.3	26.5	6.0	1.2
Item #12 “I feel nervous”	34.9	47	14.5	3.6
Item #14 “I feel indecisive”	53.7	23.2	19.5	3.7

## Data Availability

Written informed consent was obtained from the subject(s) in order to publish this paper.

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
