# Peer review of "Vaccine Hesitancy and the Green Digital Pass: A Study on Adherence to the Italian COVID-19 Vaccination Campaign"

_ijerph, 2022, doi:10.3390/ijerph19052970_

Round 1

Reviewer 1 Report

The paper sets out to analyze the motivations behind the decision not to get vaccinated against COVID-19 among Italian citizens in July 2021, after the obligatory vaccination certificate was introduced.

To this end, patients in a vaccination centre in Salerno were interviewed, and in parallel a qualitative analysis was conducted of the articles published on the daily paper Il Mattino to identify the most frequent arguments used by the press to convince those who did not want to get vaccinated.

The article shows a number of both theoretical and methodological weaknesses.

The section with the results mentions only the answer to the question on whether the respondents got vaccinated when they were called by the regional platform, reporting that 19.3% chose instead to get vaccinated after the introduction of the vaccination certificate.

No mention is made of the motivations behind the previous choice not to get vaccinated or to delay vaccination, as in contrast one would have expected in accordance with the objective of the study, as stated by the authors.

The cited literature is limited.

The analysis of the frequency of the words used by the press in connection with the vaccination certificate is inconclusive.

On the whole, the methodology is very weak, if not outright questionable.

The conclusions end up not actually focusing on the objective stated at the beginning, but rather deal with the ways to encforce or induce a certain behaviour in the population (nudges, level of confidence).

The authors claim that “Furthermore, it appears from our study that, as opposed to health anxiety or altruism, it was the fear of further social limitation that led to a reduction in vaccine hesitancy” (lines 200-201) based on the analysis of the newspaper articles that were examined. Since the press largely presented the vaccination certificate as a way to regain one’s social life, one would therefore assume that the reduction in vaccination hesitancy is due to this factor. This conclusion is not warranted in any way by the data presented in the article, and several confounding variables that were not isolated may have influenced the choices made by the individuals concerned.

If the authors’ intention had been to tackle the role of communication in the reduction of vaccination hesitancy, they should have attempted to detect its impact, and not posit a cause-effect mechanism between what the press convey and the choices of individuals, without actually measuring the breadth and relevance of the pheonomena involved.

Author Response

Comments and Suggestions for Authors

The paper sets out to analyze the motivations behind the decision not to get vaccinated against COVID-19 among Italian citizens in July 2021, after the obligatory vaccination certificate was introduced.

To this end, patients in a vaccination centre in Salerno were interviewed, and in parallel a qualitative analysis was conducted of the articles published on the daily paper Il Mattino to identify the most frequent arguments used by the press to convince those who did not want to get vaccinated.

The article shows a number of both theoretical and methodological weaknesses.

1- The section with the results mentions only the answer to the question on whether the respondents got vaccinated when they were called by the regional platform, reporting that 19.3% chose instead to get vaccinated after the introduction of the vaccination certificate. No mention is made of the motivations behind the previous choice not to get vaccinated or to delay vaccination, as in contrast one would have expected in accordance with the objective of the study, as stated by the authors.

Authors: The authors thank you for this important feedback. In fact, in the "Result" paragraph we reported the frequencies of responses to all items relating to the COVID-19 experience, in which the perception of risk and the reasons underlying the adhesion or rejection of the vaccine emerged.

2- The cited literature is limited.

Authors: The authors have expanded the bibliography as changes have been made in the paragraphs "Introduction", "Discussion" and "Conclusions"

3- The analysis of the frequency of the words used by the press in connection with the vaccination certificate is inconclusive.

Authors: The Authors appreciated this statement, in fact in order not to make errors of form and content they decided to eliminate the textual analysis of the contents obtained from the newspaper articles, but to highlight the results obtained from the analysis of the participants' responses.

4- On the whole, the methodology is very weak, if not outright questionable.

Authors: The paragraph "Materials and method" has been added with the related subpargraphs in which we have detailed the methodology of our survey.

5- The conclusions end up not actually focusing on the objective stated at the beginning, but rather deal with the ways to encforce or induce a certain behaviour in the population (nudges, level of confidence).

The authors claim that “Furthermore, it appears from our study that, as opposed to health anxiety or altruism, it was the fear of further social limitation that led to a reduction in vaccine hesitancy” (lines 200-201) based on the analysis of the newspaper articles that were examined. Since the press largely presented the vaccination certificate as a way to regain one’s social life, one would therefore assume that the reduction in vaccination hesitancy is due to this factor. This conclusion is not warranted in any way by the data presented in the article, and several confounding variables that were not isolated may have influenced the choices made by the individuals concerned.

If the authors’ intention had been to tackle the role of communication in the reduction of vaccination hesitancy, they should have attempted to detect its impact, and not posit a cause-effect mechanism between what the press convey and the choices of individuals, without actually measuring the breadth and relevance of the pheonomena involved.

Authors: On the basis of the precious indications of the Reviewer, we have reformulated the paragraph "Discussion" and "Conclusions" eliminating the hypotheses obtained from the press and discussing the results obtained from our questionnaire. The limitations present in our study were also added.

Reviewer 2 Report

This report addresses the hesitancy to receive a corona virus vaccine in relation to the implementation of governmental instructions regarding the need to be vaccinated for a number of professional workers and for access to certain places. The approach in this study was on one hand by interviewing people at a vaccination center; and on the other hand by searching articles in a main newspaper in the area of Campania, located in the south of Italy around the city of Naples. There was a drop in individuals invited for a first dose before implementation of this Green Pass, with a resumption in reservations for a vaccination after the implementation of the Green Pass. The assessment of the coverage in the local press showed that the presentation of the GreenPass mainly addressed the need to be vaccinated in relation to the impact for social life, and not the impact on health.

This report is interesting and in accord with the expectations of a mandatory status of vaccination for many actions in life, after similar passes (or QR codes) have been introduced in many (European) countries. However, the manuscript needs a substantial revision.

First, a questionnaire comprising 28 items is described, out of which 2 were used in the present study. Th details of this questionnaire need to be given, and also why 26 items were included but not used. The way in which the respondents were invited need to be described, including but not limited to the introduction/invitation, location, bias created by the person/place, and processing of data. Also it seems that the interviews were done by vaccinators, and respondents were individuals hesitant to receive a vaccination. The selection of respondents is not described. Hence, section 2.1.2 needs much more detail.

Second, the results need to be presented in much more detail, preferably by using tables in presentation. In its present form, it is almost impossible to get the data from the presentation in unclear text.

Then, the section on “conclusions” is rather a discussion. There are no interpretations nor conclusions in the paragraph.

Finally, there should be much more clarity in phrasing the text. Many sentences are just statements without a context. Hence, the manuscript is quite difficult to read.

Author Response

Comments and Suggestions for Authors

This report addresses the hesitancy to receive a corona virus vaccine in relation to the implementation of governmental instructions regarding the need to be vaccinated for a number of professional workers and for access to certain places. The approach in this study was on one hand by interviewing people at a vaccination center; and on the other hand by searching articles in a main newspaper in the area of Campania, located in the south of Italy around the city of Naples. There was a drop in individuals invited for a first dose before implementation of this Green Pass, with a resumption in reservations for a vaccination after the implementation of the Green Pass. The assessment of the coverage in the local press showed that the presentation of the GreenPass mainly addressed the need to be vaccinated in relation to the impact for social life, and not the impact on health.

This report is interesting and in accord with the expectations of a mandatory status of vaccination for many actions in life, after similar passes (or QR codes) have been introduced in many (European) countries. However, the manuscript needs a substantial revision.

1- First, a questionnaire comprising 28 items is described, out of which 2 were used in the present study. Th details of this questionnaire need to be given, and also why 26 items were included but not used. The way in which the respondents were invited need to be described, including but not limited to the introduction/invitation, location, bias created by the person/place, and processing of data. Also it seems that the interviews were done by vaccinators, and respondents were individuals hesitant to receive a vaccination. The selection of respondents is not described. Hence, section 2.1.2 needs much more detail.

Authors: We have inserted the paragraph "Materials and methods" in which we have described in detail the characteristics of the participants, the questionnaire used, the administration and recruitment procedure. All the items of the questionnaire were used and in the "Results" section the response frequencies to each item were further described.

2- Second, the results need to be presented in much more detail, preferably by using tables in presentation. In its present form, it is almost impossible to get the data from the presentation in unclear text.

Authors: The results have been better described and descriptive tables have been inserted.

3- Then, the section on “conclusions” is rather a discussion. There are no interpretations nor conclusions in the paragraph.

Authors: The "Discussion" paragraph has been added while the "Conclusions" paragraph has been clarified in form and content. The limitations of our study were also included.

4- Finally, there should be much more clarity in phrasing the text. Many sentences are just statements without a context. Hence, the manuscript is quite difficult to read.

Authors:  We have improved the form and content of the entire text. Linguistic revision was done again.

Round 2

Reviewer 2 Report

The authors have carefully considered the comments on the original manuscript, and have made manyny revisions in accordance with these comments. The response to the comments is appreciated.